# Post Hoc Explanations of Language Models Can Improve Language Models

Satyapriya Krishna[1], Jiaqi Ma[2], Dylan Slack[3], Asma Ghandeharioun[4], Sameer Singh[3], and Himabindu Lakkaraju[1]

[1]Harvard University
[2]University of Illinois Urbana-Champaign
[3]University of California, Irvine
[4]Google Inc
skrishna@g.harvard.edu

## Abstract

Large Language Models (LLMs) have demonstrated remarkable capabilities in performing complex tasks. Moreover, recent research has shown that incorporating human-annotated rationales (e.g., Chain-of-Thought prompting) during in-context learning can significantly enhance the performance of these models, particularly on tasks that require reasoning capabilities. However, incorporating such rationales poses challenges in terms of scalability as this requires a high degree of human involvement. In this work, we present a novel framework, Amplifying Model Performance by Leveraging In-Context Learning with Post Hoc Explanations (AMPLIFY), which addresses the aforementioned challenges by automating the process of rationale generation. To this end, we leverage post hoc explanation methods which output attribution scores (explanations) capturing the influence of each of the input features on model predictions. More specifically, we construct automated natural language rationales that embed insights from post hoc explanations to provide corrective signals to LLMs. Extensive experimentation with real-world datasets demonstrates that our framework, AMPLIFY, leads to prediction accuracy improvements of about 10-25% over a wide range of tasks, including those where prior approaches which rely on human-annotated rationales such as Chain-of-Thought prompting fall short. Our work makes one of the first attempts at highlighting the potential of post hoc explanations as valuable tools for enhancing the effectiveness of LLMs. Furthermore, we conduct additional empirical analyses and ablation studies to demonstrate the impact of each of the components of AMPLIFY, which, in turn, lead to critical insights for refining in-context learning.

## 1   Introduction

In recent years, Large Language Models (LLMs) [3] have ushered in a new era for machine learning research. These models are exhibiting emergent capabilities that enable them to excel at complex tasks that involve sophisticated capabilities such as reasoning and language understanding [34, 5]. Moreover, these models do not only exhibit remarkable performance on tasks they were trained for but also quickly adapt to other novel and complex tasks. This is made possible through a mechanism known as *in-context learning*, which allows these models to learn from a limited number of input and label pairs, commonly referred to as few-shot prompts[8], provided during test time. Prior research has also demonstrated that the performance of these models on sophisticated reasoning tasks can be significantly improved by presenting them with human-annotated rationales alongside

37th Conference on Neural Information Processing Systems (NeurIPS 2023).

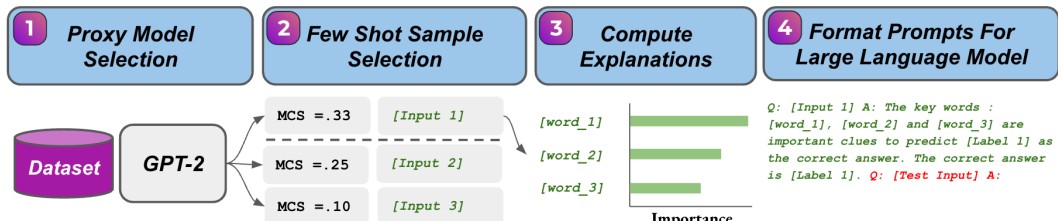

**Figure 1:** The AMPLIFY framework consists of four steps aimed at improving the performance of LLMs. (1) We select a proxy model, such as GPT-2 or BERT, which is significantly smaller in size compared to the LLMs and for which it is computationally feasible to generate post hoc explanations. (2) By leveraging the validation set, we identify samples that were misclassified by the LLM. Subsequently, we select the samples that the proxy model exhibits the highest level of confidence in misclassifying. (3) We then use explainability techniques to compute explanations for the selected samples with respect to their ground truth labels. (4) We construct the few-shot prompt for LLM using the samples selected and their corresponding explanations to feed as input to LLM for prediction.

input/label pairs during test time [35, 15]. While incorporating such human-annotated rationales has contributed to enhancing model performance, it is not a scalable approach as it involves a lot of human intervention, thus, limiting its applicability to the ever-expanding range of tasks that are being handled by LLMs [29, 38]. Additionally, prompting these models with human-annotated rationales is not always effective, and may lead to drops in performance on certain tasks requiring sophisticated language understanding as demonstrated by prior works [31]. This could be due to the fact that the seemingly innocuous biases introduced by human-annotated rationales do not necessarily align with the performance goals [33].

To address the aforementioned challenges, we propose a novel framework Amplifying Model Performance by Leveraging In-Context Learning with Post Hoc Explanations (AMPLIFY) which can automatically generate rationales to improve the performance of LLMs on tasks involving sophisticated reasoning and language understanding. To this end, we leverage post hoc explanation methods which output attribution scores that capture the influence of each of the input features on model predictions. More specifically, our framework constructs natural language rationales that embed insights from post hoc explanations to provide corrective signals to LLMs. For example, when a LLM makes an error on an instance, our framework enables the model to correct itself by first identifying the top keywords by computing post hoc explanations (e.g., gradients) of the model with respect to the given instance and its true label, and then prompting the model to pay attention to the top keywords identified, thus, amplifying this signal. To illustrate, let us consider a sentiment classification task where a LLM is assessing the sentiment associated with the sentence *"RRR movie has a great story and amazing visuals."*. If the model is incorrect in its assessment, it can be provided with a corrective input such as "The keywords 'great' and 'amazing' are important cues in predicting the sentiment of this sentence" where the keywords themselves are automatically identified by a post hoc explanation method. While post hoc explanations have generally been considered valuable tools for deepening our understanding of model behavior [6] and for identifying root causes of errors made by ML models [10, 11], our work is the first to explore their utility in improving the performance of LLMs.

While the use of post hoc explanation methods to rectify LLMs' behavior eliminates the need for human intervention, it encounters two significant challenges: First, calculating gradients (and hence post hoc explanations) for LLMs with several billions of parameters is computationally intensive. Second, many LLMs operate as black boxes, depriving end users of access to gradients or other internal model details. To mitigate these issues, we compute post hoc explanations for a proxy model (e.g. GPT-2, BERT, etc.) that is considerably smaller than a large language model with billions of parameters. We then incorporate these explanations into input prompts for larger language models (e.g. GPT-3, GPT-3.5, etc.). This approach not only improves efficiency and feasibility (as we are now calculating gradients for models with 100-200 million parameters, instead of those with 100 billion parameters) but also takes advantage of the accessibility of smaller models that are open sourced. In summary, our AMPLIFY framework follows a four-step approach: (1) Select a proxy model for which post hoc explanation generation is computationally viable; (2) Identify samples that are most likely to provide corrective signals to LLMs; (3) Compute post hoc explanations for the samples identified

in the previous step; (4) Construct a few-shot prompt using the samples chosen in step (2), their true labels, and the post hoc explanations obtained in step 3 as rationales. This prompt is then provided as input to the LLM at test time.

Our findings demonstrate that `AMPLIFY` leads to performance improvements of about 10-25% across a wide range of tasks, including those where previously considered prompting techniques such as Chain-of-Thought prompting which rely on human-annotated explanations, fall short. This highlights the potential of post hoc explanation methods as valuable tools for enhancing the effectiveness of LLMs. Furthermore, we conduct an extensive empirical analysis to examine the impact of each step of our framework `AMPLIFY`. This allows for a better understanding of the change in LLM performance with different choices of proxy model (step 1), selection strategy (step 2), post hoc explanation method (step 3), and rationale templates (step 4). Thus, we offer critical insights for refining in-context learning while addressing the limitations posed by methods dependent on human-annotated rationales.

## 2 Related Works

**In-context Learning.**  Over the past decade, numerous language models have been developed to excel at a wide range of complex predictive tasks [34]. This is accomplished by training and fine-tuning language models on datasets associated with various tasks. While these advancements have led to highly effective language models for numerous tasks, they have also increased the models' parameter sizes and the computational costs for additional fine-tuning on new tasks. To address this issue, recent research has demonstrated that modern language models can learn new tasks in-context, which allows the model to perform well on new tasks by simply providing a few task samples in the prompt [16]. This method of learning contrasts with the conventional fine-tuning process, which incurs additional computational costs [16]. This in-context learning ability is even more pronounced in extremely large language models (>100 billion parameters), where it is also referred to as an "emergent ability" [34]. These findings have garnered significant attention, and various approaches have been proposed to enhance in-context learning by incorporating additional signals into the prompt [15]. The current state-of-the-art among such approaches is the Chain-of-Thought (CoT) technique [35] which augments prompts with human-annotated rationales comprising of step-by-step instructions on how to perform a new task. This method has substantially improved language models' capacity to tackle highly challenging tasks that involve sophisticated reasoning. However, this method relies heavily on human annotations and is therefore not very scalable. Further, prior works have demonstrate that this method also leads to poor performance in certain kinds of reasonings tasks[31]. These limitations persist, and there are largely no solutions for them yet. In this work, we propose an approach that demonstrates a lot of promise in alleviating the aforementioned issues.

**Post Hoc Explanations.**  As language models have become more capable and complex, understanding their behavior and the rationale behind their predictions has grown increasingly challenging [9]. To understand the predictions made by these black boxes, various methods have been developed to provide explanations in the form of feature attributions which capture the influence of each input feature on a given model prediction. These methods are known as post hoc explanation methods [19]. Post hoc explanation methods can be broadly classified into two primary categories: (1) perturbation-based methods and (2) gradient-based methods. Perturbation-based methods involve creating an interpretable approximation of the original black-box model using perturbations of input samples. Notable examples of these methods include LIME [23], SHAP [18], Occlusion [37], and others. In addition, gradient-based methods such as SmoothGrad and Integrated Gradients [28, 30] calculate model gradients with respect to input features to determine the sensitivity of the model's output to each feature. However, to the best of our knowledge, the utility of these explanation methods has not been studied in the context of LLMs. Our work makes the first attempt at exploring the utility of these methods in improving the performance of LLMs.

## 3 Our Framework `AMPLIFY`

In this section, we describe our framework Amplifying Model Performance by Leveraging In-Context Learning with Post Hoc Explanations (`AMPLIFY`) in detail. Recall that the main goal of our framework is to eliminate the need for human-annotated rationales, and to instead generate automated rationales which can, in turn, provide corrective signals to improve the performance of LLMs on

sophisticated reasoning and language understanding tasks. To this end, our framework leverages post hoc explanations to construct such rationales. More specifically, our framework adopts the following four-step approach: (1) Select a proxy model for which post hoc explanation generation is computationally viable; (2) Identify samples that are most likely to provide corrective signals to LLMs; (3) Compute post hoc explanations for the samples identified in the previous step; (4) Construct a few-shot prompt using the samples chosen in step (2), their true labels, and the post hoc explanations (rationales) obtained in step 3. This prompt is then provided as input to the LLM at test time. We discuss each of these steps in more detail below.

**Step (1): Proxy Model Selection.** In this step, we choose a proxy model, typically one that is substantially smaller in size compared to LLMs with billions of parameters, so that generating post hoc explanations is computationally viable. Further, we consider a couple of strategies when selecting a suitable proxy model: (i) Use a pre-trained model such as GPT-2, BERT, etc., which has been shown to perform quite well on several tasks and is thousands of times smaller than LLMs (GPT-3, Bloom, etc.) or (ii) Fine-tune or pre-train a smaller language model from scratch on the target task. The major difference between the two strategies is that the first one requires no additional computational cost as we directly use a pre-trained (potentially open-sourced) model. We test both proxy model selection strategies to discern performance variations between them. Lastly, it is important to note that proxy models of the size we select in this step (e.g., GPT-2, BERT etc.) do not exhibit complex reasoning abilities [34]. Consequently, they do not perform well on reasoning tasks by themselves [29]. However, our analyses (more details in Section 4) demonstrate that such smaller models can be used to improve the reasoning capabilities and task performance of LLMs.

**Step (2): Few-shot Sample Selection.** The goal of this step is to identify samples i.e., (input, label) pairs that are most likely to provide corrective signals to the LLM. To this end, we first identify instances from the validation set that are misclassified by the LLM. We then rank these instances using a metric we introduce called the Misclassification Confidence Score (MCS). Formally, $\text{MCS}(\mathbf{x}) = \mathbf{f}(\mathbf{x})_{\mathbf{y}} - \mathbf{f}(\mathbf{x})_{\hat{\mathbf{y}}}$. Here, $\mathbf{x} \in \mathbb{R}^N$ represents the input sequence containing N tokens, $f : \mathbb{R}^N \to \mathbb{R}^{|\mathcal{L}|}$ is the fine-tuned language model that produces class probability scores for each label in the label set $\mathcal{L}$, $f(\mathbf{x})_{\mathbf{y}}$ represents the class probability score for the incorrect label ($y$) predicted by the model, and $f(\mathbf{x})_{\hat{\mathbf{y}}}$ represents the class probability score for the ground truth label ($\hat{y}$). The samples that exhibit the highest MCS represent the most egregious misclassifications. By incorporating these samples and their corresponding corrective rationales into the few-shot prompt, the LLM is likely to receive strong supervision to avoid similar misclassifications. In summary, this step results in $s$ samples of (input ($\mathbf{x}$), label ($\hat{y}$)) pairs, filtered from the validation set, that are likely to carry the most useful corrective signals to assist LLMs.

**Step (3): Rationale Generation.** In this step, we compute post hoc explanations for each sample obtained from the previous step. Specifically, for each sample, we use a post hoc explanation method with the (input, label) pair, along with the proxy model, which then calculates the attribution scores for each token in the input sentence. These attribution scores, associated with each token, indicate the contribution each token in the input sentence makes towards the proxy model's prediction of the provided label. We then compute attribution scores for each word by averaging the attributions obtained for each token in that word. Finally, we filter the top-$k$ words with the highest attribution scores. As a result, this step outputs a set of $k$ words for each sample selected in the previous step. The most commonly used post hoc explanation methods for computing attribution scores of input tokens are based on gradient computations [19]. That is, the attribution for the token $\mathbf{x_i}$ in input $\mathbf{x} \in \mathbb{R}^N$ is computed as $\frac{\partial f(\mathbf{x})_{\hat{y}}}{\partial \mathbf{x_i}}$, as is the case with Vanilla Gradients [25]. We experiment with several other post hoc explanation methods discussed in more detail in the experiment section.

**Step (4): Prompt Design for LLMs.** In the final step, we proceed to construct the corrective rationale for each selected sample using the template: *"The key words: $word_1$, $word_2$, ...and $word_k$ are important clues to predict [Label] as the correct answer."* In this template, "[ $word_1$, $word_2$..., and $word_k$ ]" refers to the top-$k$ most important words in the input sentence for the true label, which were obtained from the previous step. Using the few-shot template *[Input][Rationale][Label]*, we construct the $s$-shot prompt as *"[Input$_1$][Rationale$_1$][Label$_1$]...[Input$_s$][Rationale$_s$][Label$_s$]"*, which is simply the concatenation of *[Input][Rationale][Label]* for each selected sample. This constructed prompt is then combined with the input of the test sample to form the final input prompt

for the LLMs, enabling them to make predictions for the samples in the test set. This process is illustrated in the last step of Figure 1.

## 4 Experimental Evaluation

In this section, we discuss our empirical evaluation in detail. First, we describe our experiment setup and provide details about the datasets and tasks we experiment with. Next, we evaluate the effectiveness of our framework in improving task performance of LLMs on a variety of real-world tasks. Lastly, we examine the impact of each step of our framework AMPLIFY by conducting rigorous ablation studies.

**Datasets.** We evaluate our framework AMPLIFY on some of the popular datasets from the Big-Bench-Hard[29] benchmark. More specifically, we experiment with: (1) The *Snarks*[29] dataset which gauges a model's proficiency in discerning sarcastic sentences from a selection of alternatives; (2) The *Causal Judgment*[29] dataset, designed to evaluate a model's ability in accurately deducing the causative factors of an event from a detailed summary; (3) The *Ruin Names*[29] task, which involves the identification of comical modifications to artist or movie names; (4) The *Formal Fallacies*[29] task, where machine learning models are put to the test to distinguish between logically sound arguments and those with logical discrepancies; (5) The *Salient Translation Error Detection*[29] task, engineered to train models in identifying one out of six predetermined translation errors given a pair of translations; (6) The *CommonsenseQA* [32] dataset, a multiple-choice question answering platform that necessitates a wide variety of commonsense knowledge for accurately determining the correct responses; (7) Lastly, the *Coin Flip* [35] dataset, a synthetically generated dataset used for assessing the symbolic reasoning capacity of LLMs.

**Large Language Models.** Our methodology is assessed in comparison to baseline approaches on two LLMs. First, GPT-3 [4], a language model with 175 billion parameters, demonstrates robust performance across a range of natural language processing tasks without the need for explicit training or fine-tuning. Second, GPT-3.5 [1] is a series of models that were trained on a mix of text and code data before the end of the fourth quarter in 2021. These models, expressly crafted for chat applications, function as an advanced version of InstructGPT [20].

**Post Hoc Explanation Techniques.** In this study, we use three widely adopted post hoc explanation methods to generate explanations that are later incorporated as rationales into prompts for in-context learning. These methods include Vanilla Gradients [25], Gradient x Input [24], and contrastive explanations [36]. Vanilla Gradients [25] calculates feature attributions by computing the norm of the gradients of model output with respect to each token's embedding. Gradient x Input derives attribution scores by taking the product of gradient and input embedding. Finally, contrastive gradients determine attribution scores by subtracting the gradients with respect to the model prediction token from those associated with the truth label. We apply these explanation techniques to two proxy models for the generation of corrective rationales in step 3 of AMPLIFY: GPT-2 ($\sim$125 Mn parameters)[21] and BERT ($\sim$110 Mn parameters) [7].

**Baseline Methods.** In our experiments, we evaluate the performance of AMPLIFY in comparison to two alternative approaches, namely Answer-Only (AO) prompts [29] and Chain-of-Thought (CoT) [35]. AO prompting represents the standard few-shot prompting technique, in which the input prompt consists of a few (input, label) pairs and the test input sentence, followed by an answer delimiter ('A:') for the LLM to generate the response. Chain-of-Thought (CoT), on the other hand, is the current state-of-the-art method that enhances the input prompt by including human-annotated rationales for each few-shot example. The LLM is then expected to generate a rationale followed by an answer.

**Implementation Details.** In our experiments, we implemented the AO and CoT baselines using the same methodology as described in their respective works. For CoT, we directly used the provided rationales from the original work for the corresponding datasets [35]. In the case of AMPLIFY, we employed GPT-2[22] fine-tuned for the target task as the proxy model for step 1, unless stated otherwise. We utilized a rationale template with $k = 5$, which is of the form: *"The key words: word$_1$, word$_2$, ...and word$_5$ are important clues to predict [ground truth label] as the correct answer"*. These keywords *"word$_1$, word$_2$, ...and word$_5$"* were chosen based on the highest attribution scores obtained

**Table 1:** Few-shot prompting performance of several large language models on the seven datasets. AO: standard "answer-only" prompting. CoT: chain-of-thought prompting. Best model performance is in bold. The LLMs we experimented with are GPT-3 and GPT-3.5. The recorded performance in this table represents the percentage of test samples for which the LLM accurately predicted the true label.

| | [29, 35] | | Human-Rater [38] | | GPT-3 | | | GPT-3.5 | | |
|---|---|---|---|---|---|---|---|---|---|---|
| Experiment Tasks | Random | SOTA | Avg. | Max | AO | CoT | AMPLIFY | AO | CoT | AMPLIFY |
| Snarks | 50.0 | 71.3 | 76.7 | 100 | 52.7 | 61.1 | 80.5 | 75.0 | 69.4 | **91.6** |
| Causal Judgment | 50.0 | 62.1 | 69.6 | 100 | 55.2 | 55.2 | 60.5 | 57.8 | 63.1 | **76.3** |
| Ruin Names | 25.0 | 72.8 | 77.7 | 100 | 64.0 | 62.9 | **78.6** | 69.6 | 62.9 | 77.5 |
| Formal Fallacies | 25.0 | 52.2 | 90.8 | 100 | 53.6 | 50.8 | **60.1** | 51.4 | 54.6 | 59.9 |
| Salient Translation Error Detection | 16.7 | 31.9 | 36.7 | 80.0 | 48.2 | 50.2 | 51.7 | 43.2 | 54.7 | **60.8** |
| CommonsenseQA | 20.0 | 80.0 | 90.0 | 100 | 69.3 | 72.6 | 73.5 | 75.7 | 75.2 | **77.9** |
| Coin Flip (OOD) | - | - | - | - | 54.7 | 63.3 | **65.7** | 52.9 | 61.0 | 65.3 |
| All Tasks *(avg)* | 31.1 | 61.7 | 73.5 | 96.6 | 56.8 | 58.0 | 67.2 | 60.8 | 62.9 | **72.7** |

**Table 2:** Few-shot prompting performance of multiple LLMs on the seven datasets when post hoc explanations, which form the rationale in the prompt constructed during step 4 of AMPLIFY, are generated using models with varying degrees of fine-tuning of the proxy model (GPT-2 in this case). Here, "E" represents the number of epochs the proxy model was fine-tuned. "E = 0" indicates that the proxy model was used to generate post hoc explanations without any fine-tuning. The recorded performance in this table represents the percentage of test samples for which the LLM accurately predicted the true label.

| | GPT-3 | | | GPT-3.5 | | |
|---|---|---|---|---|---|---|
| Experiment Tasks | E = 0 | E = 10 | E = 200 | E = 0 | E = 10 | E = 200 |
| Snarks | 77.7 | 80.5 | 80.5 | 88.8 | 88.8 | **91.6** |
| Causal Judgment | 55.2 | 57.8 | 60.5 | 71.0 | 73.6 | **76.3** |
| Ruin Names | 74.1 | 75.2 | **78.6** | 65.1 | 67.4 | 77.5 |
| Formal Fallacies | 53.7 | 56.9 | **60.1** | 48.3 | 51.6 | 59.8 |
| Salient Translation Error Detection | 49.7 | 51.2 | 51.7 | 57.7 | 60.8 | **60.8** |
| CommonsenseQA | 69.1 | 72.6 | 73.5 | 71.9 | 75.8 | **77.9** |
| Coin Flip (OOD) | 56.4 | 60.8 | **65.7** | 55.4 | 61.4 | 65.3 |
| All Tasks *(avg)* | 62.2 | 65.0 | 67.2 | 65.4 | 68.4 | **72.7** |

from the post hoc explanation computed in step 3. To compute these attribution scores, we used Gradient x Input as the default post hoc explanation method for generating explanations.

## 4.1 Empirical Analysis

**Overall Task Performance.** We demonstrate the effectiveness of AMPLIFY by comparing the prediction accuracy of LLMs using prompts generated by AMPLIFY against baselines, i.e., Answer-Only (AO) prompts and Chain-of-Thought (CoT) prompts. Table 1 displays the results of GPT-3 and GPT-3.5 across all datasets. We observe that incorporating rationales generated by our approach leads to a substantial improvement in accuracy compared to both standard Answer-Only (AO) prompts and Chain-of-Thought (CoT) prompts. Specifically, GPT-3.5 achieves state-of-the-art performance on the Snarks dataset with a 91.6% accuracy in identifying the correct option; this is 16% better than standard answer-only prompting and over 20% better than CoT. Similar trends were observed for Causal Judgment, where our method delivered the best performance of 76.3%, significantly surpassing CoT (63.1%) and AO (57.8%). When using GPT-3, our approach attained the highest performance in Ruin Names (78.6%), a trend also evident in the case of Formal Fallacies. Finally, our method achieved the top performance with the GPT-3.5, registering an accuracy of 60.8% for the Salient Translation Error Detection task. In the scenarios of commonsense reasoning (CommonsenseQA) and symbolic reasoning (Coin Flip) tasks, we noticed consistent trends, with AMPLIFY recording the highest performance. In the next set of analyses, we examine the effects of different steps in AMPLIFY on the performance of LLMs.

**Impact of Proxy Model Selection on LLM Performance.** In the following analysis, we investigate how the choices made at each step of AMPLIFY affect the performance of LLM on the seven tasks. We begin with step 1, which involves selecting a proxy model for sample selection (step 2) and computing post hoc explanations(step 3). In the experiments conducted to calculate the overall model performance, as shown in Table 1, we utilized a finetuned GPT-2 model for the target task as

**Table 3:** Few-shot prompting performance of various large language models on the seven datasets is analyzed based on different selection strategies used for choosing samples during prompt design of LLMs, specifically in step 2 of Figure 1. The "Random" selection refers to randomly chosen samples. "L-MCS " signifies the selection of samples with the lowest Misclassification Confidence Score (MCS). "H-MCS " represents the selection strategy of choosing samples with the Misclassification Confidence Score (MCS) for prompt design. "F-Exp" indicates the selection strategy of choosing samples with the most faithful explanations for LLM prompt. The recorded performance in this table represents the percentage of test samples for which the LLM accurately predicted the true label.

| Experiment Tasks | GPT-3 | | | | GPT-3.5 | | | |
|---|---|---|---|---|---|---|---|---|
| | Random | L-MCS | H-MCS | F-Exp | Random | L-MCS | H-MCS | F-Exp |
| Snarks | 69.4 | 80.5 | 80.5 | 77.7 | 69.4 | 88.8 | **91.6** | 88.8 |
| Causal Judgment | 57.8 | 60.5 | 60.5 | 57.8 | 68.4 | 73.6 | **76.3** | 71.0 |
| Ruin Names | 65.1 | 77.5 | **78.6** | 74.1 | 66.2 | 77.5 | 77.5 | 73.0 |
| Formal Fallacies | 52.3 | 57.9 | **60.1** | 59.7 | 46.7 | 51.5 | 59.9 | 58.6 |
| Salient Translation Error Detection | 48.7 | 50.2 | 51.7 | 51.7 | 53.2 | 59.2 | **60.8** | 58.7 |
| CommonsenseQA | 67.6 | 71.5 | 73.5 | 70.9 | 72.9 | 76.6 | **77.9** | 77.2 |
| Coin Flip (OOD) | 54.7 | 60.1 | **65.7** | 58.5 | 57.8 | 61.6 | 65.3 | 61.1 |
| All Tasks *(avg)* | 59.3 | 65.4 | 67.2 | 64.3 | 62.0 | 69.8 | **72.7** | 69.7 |

**Table 4:** The table presents a performance comparison for when prompt is created using explanations generated by four different post hoc explanation methods. Grad: Vanilla Gradient Method, Grad × Inp : Gradient x Input Method , C-Grad and C-Grad×Inp are contrastive version of Vanilla gradient and Gradient x Input. The recorded performance in this table represents the percentage of test samples for which the LLM accurately predicted the true label.

| Experiment Tasks | GPT-3 | | | | GPT-3.5 | | | |
|---|---|---|---|---|---|---|---|---|
| | Grad | Grad×Inp | C-Grad | C-Grad×Inp | Grad | Grad×Inp | C-Grad | C-Grad×Inp |
| Snarks | 77.7 | 80.5 | 80.5 | 86.1 | 88.8 | **91.6** | 88.8 | 91.6 |
| Causal Judgment | 60.5 | 60.5 | 57.8 | 60.5 | 71.0 | **76.3** | 71.0 | 73.6 |
| Ruin Names | 71.9 | **78.6** | 75.2 | 77.5 | 65.1 | 77.5 | 73.0 | 74.1 |
| Formal Fallacies | 59.7 | **60.1** | 59.7 | 58.6 | 59.9 | 59.9 | 59.4 | 57.6 |
| Salient Translation Error Detection | 49.7 | 51.7 | 51.7 | 50.7 | 59.7 | **60.8** | 60.8 | 60.8 |
| CommonsenseQA | 72.1 | 73.5 | 72.9 | 73.0 | 73.7 | **77.9** | 75.5 | 77.9 |
| Coin Flip (OOD) | 62.9 | 64.1 | 62.6 | **65.7** | 62.6 | 63.9 | 62.4 | 65.3 |
| All Tasks *(avg)* | 64.9 | 67.0 | 65.7 | 67.3 | 68.6 | **72.5** | 70.1 | 71.5 |

the proxy model. While using GPT-2 for finetuning is computationally cheaper compared to other LLMs, it is still expensive to finetune a model with more than 100 million parameters. Therefore, we examined the performance of LLMs based on the amount of fine-tuning, measured in terms of the number of epochs (E). This analysis aimed to understand the impact of finetuning on improving model performance. Table 2 presents the model performance scores of LLMs when the proxy model is GPT-2 without any fine-tuning on the target task (E=0), with minor fine-tuning (E=10), and when GPT-2 has achieved its best performance at epoch E=200. As depicted in Table 2, we observe that the model performance of LLMs with GPT-2 (E=0) is already quite close to the best performance achieved when GPT-2 is finetuned to saturation (E=200) for most datasets. Specifically, the performance of GPT-3.5 for Snarks with GPT-2 (E=0) is 88.8%, which is significantly better than the best performance of CoT shown in Table 1. This pattern is also evident in the case of Causal Judgment, Salient Translation Error Detection, and Ruin Names. There are two primary reasons for this behavior. Firstly, GPT-2 possesses sufficient language understanding capabilities to provide useful post hoc explanations that lead to accurate predictions. Secondly, most of the tasks where GPT-2 (E=0) achieved the best or near-best performance have very limited training data, which might not be sufficient for GPT-2 to learn anything beyond what it has already acquired during pre-training. This observation is further supported by the findings presented in Table 5 in Appendix A, where the accuracy improvements for most datasets are not substantial. These findings suggest that an additional step of fine-tuning a pre-trained model may not always be necessary when selecting a proxy model in step 1 of AMPLIFY, thereby reducing computational costs even further. Lastly, we observe similar trends when BERT is chosen as the proxy model, and the detailed results are presented in Appendix A.4.

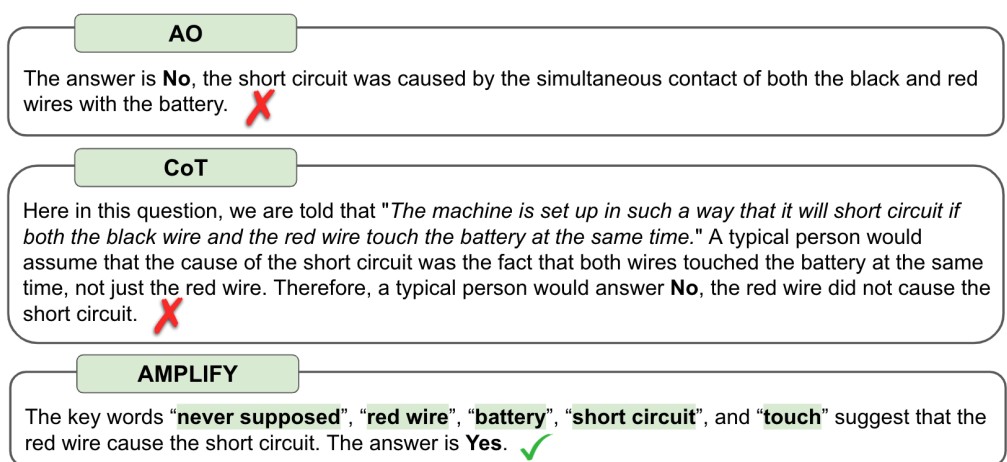

**Figure 2:** This image exemplifies an instance of Causal Judgment task where standard prompts and CoT produce inaccurate responses. The CoT response fails to take into account that the red wire should not make contact with the battery, which caused the short circuit. In contrast, the response generated by AMPLIFY emphasizes this crucial detail. Please note that while we inject rationales in terms of k individual words, we do not restrict LLMs from generating rationales in terms of phrases or multiple words. This is why we often see LLM-generated rationales having multi-word clues, such as *"red wire," "never supposed,"* and so on.

**Impact of Selection Strategies on LLM Performance.** In this analysis, we focus on step 2 of AMPLIFY, which involves selecting samples for few-shot prompts that can provide effective corrective signals to assist LLMs in reducing misclassifications. As explained in section 3, this step in AMPLIFY includes two sub-steps: first, identifying misclassified samples by LLM on the validation set, and second, selecting samples with the highest MCS calculated with respect to proxy model. The first step is straightforward as we specifically focus on samples that were misclassified earlier by LLMs. For subsequent filtering based on confidence, we experiment with several sample selection strategies to better understand how the performance of the language model is sensitive to these different strategies. The results, in terms of language model performance scores, are shown in Table 3. Specifically, we experiment with three selection strategies in addition to the one that selects samples with the highest MCS score, referred to as "H-MCS " in Table 3: (1) "Random": randomly selecting samples without considering Misclassification Confidence Score, (2) "L-MCS": selecting samples with the lowest MCS, and (3) "F-EXP": selecting samples based on the most faithful explanations, measured by the difference in model output when the top-K features identified by the explanation are perturbed [27, 33]. Among these strategies, we find that selecting samples with the highest Misclassification Confidence Score yields the best performance, outperforming all other strategies. Random selection performs the worst across all datasets, while the performance using "L-MCS" is comparable to that of "H-MCS" selections. This suggests that samples for which the model is less confident in its predictions can still provide reasonable corrective signals in reducing misclassifications. Notably, the LLM performance with "F-EXP" sample selection is worse than "L-MCS" for most datasets (Snarks, Causal Judgment, Ruin Names, and CommonsenseQA), indicating that relying solely on faithful explanations may not be sufficient for achieving optimal performance.

**Impact of Post Hoc Explanation Method on LLM Performance.** We then examine the impact on LLM performance due to the choice of post hoc explanation used to identify top-k keywords in step 3. To investigate this, we employ four different explanation methods for step 3 of `AMPLIFY` and record the LLM performance corresponding to each post hoc explanation method choice in Table 4. Specifically, the four post hoc explanation methods used in this analysis are: (1) Vanilla Gradients [25] (Grad), (2) Gradient $\times$ Input [24] (Grad$\times$Inp), (3) Contrastive Gradient [36] (C-Grad), and (4) Contrastive Gradient $\times$ Input [36] (C-Grad$\times$Inp). Based on Table 4, we observe that the LLM performs best in general when Gradient x Input or its contrastive variant is used to generate explanations. However, we also note that there aren't drastic changes in LLM performance across different methods. For instance, GPT-3.5 performance on Snarks doesn't change much across different methods, as the accuracy remains consistently around 88.8-91.6%. This suggests that LLM performance isn't sensitive to rationales generated using different variants of gradient-based post hoc explanation methods.

**Impact of Rationale Template on LLM Performance.** Lastly, in the final step of `AMPLIFY`, we generate a few-shot prompt by combining an (input, label) pair and its corresponding set of more important words using the rationale template as *"The key words: $word_1$, $word_2$, ...and $word_5$ are crucial clues for predicting [ground truth label] as the correct answer"*. We have observed that while using a task-independent rationale template leads to improvements in performance, tailoring the rationale to the question asked in the input sentence for a given dataset also provides benefits. For example, in the case of Causal Judgment, each sample includes a generic question: *"How would a typical person answer each of the following questions about causation?"* If we utilize the rationale template as *"After observing the key words: $word_1$, $word_2$, ...and $word_5$, a typical person would respond with [label] as the correct answer"*, we notice a slight enhancement in GPT-3 performance, rising from 60.5% to 63.1%. However, we did not observe the model's sensitivity to minor changes in the template, such as punctuation variations. Further discussions on the impact of hyperparameters associated with `AMPLIFY` can be found in Appendix A.3.

**Qualitative Analysis.** In addition to quantitatively evaluating the performance of `AMPLIFY` compared to other baselines, we also qualitatively examine how LLM responses differ for certain test samples using each of the baseline prompting approaches, and compare them to the responses generated by `AMPLIFY`. Figure 2 illustrates the responses generated by GPT-3.5 for an input sample using the Standard Prompt (AO), Chain-of-Thought (CoT), and `AMPLIFY`. In this particular example, both AO and CoT yield incorrect responses, whereas `AMPLIFY` produces the correct response. Analyzing the responses reveals that CoT and AO miss an important sentence in the sample: *"The red wire is never supposed to touch the battery as it moves around inside the machine"*. Interestingly, GPT-3.5 captures this crucial information when the prompt is augmented with post hoc explanations using `AMPLIFY`. We observe similar examples for CommonsenseQA, such as *"Q: Unlike a spider and its many observers, people only have what? Answer Choices: (A) tongues (B) names (C) brains (D) feelings (E) two eyes."*. In this case, CoT incorrectly selects option (C), whereas `AMPLIFY` correctly predicts option (E). The complete response is shown in Figure 3 in the Appendix A. This error also stems from the same issue of CoT overlooking crucial information that the question pertains to eyes rather than the entire body, a nuance that `AMPLIFY` successfully captures. This demonstrates that the rationales generated by `AMPLIFY` can assist LLMs in capturing critical signals that might have otherwise been overlooked.

## 5 Conclusion

In this work, we introduce `AMPLIFY`, a novel framework aimed at improving the performance of LLMs by replacing human-annotated rationales with automated rationales obtained using post hoc explanation techniques. Our unique four-step approach leverages smaller, open-source models for efficient computation of post hoc explanations. Our framework results in performance improvements of 10-25% across diverse tasks, outperforming conventional techniques such as CoT prompting which rely on human-annotated rationales. Our findings highlight the potential of post hoc explanation methods as valuable tools for enhancing the effectiveness of LLMs.

## Acknowledgments and Disclosure of Funding

We thank Adam Pearce for helpful feedback on writing and presentation of this work. This work is supported in part by the NSF awards IIS-2008461, IIS-2040989, IIS-2238714, and research awards from Google, JP Morgan, Amazon, Harvard Data Science Initiative, and the Digital, Data, and Design (D^3) Institute at Harvard. The views expressed here are those of the authors and do not reflect the official policy or position of the funding agencies. This work was also funded in part by Hasso Plattner Institute (HPI) through the UCI-HPI fellowship, and in part by the NSF awards IIS-2008956, IIS-2046873, and IIS-2040989.

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

# A  Appendix

## A.1  Proxy Model Task Performance

**Table 5:** Proxy models performance on the target tasks with and without fine-tuning.

| | GPT-2 | | BERT | |
| --- | --- | --- | --- | --- |
| Experiment Tasks | Pre-trained | Fine-tuned | Pre-trained | Fine-tuned |
| Snarks | 38.8 | 47.2 | 30.5 | 38.8 |
| Causal Judgment | 44.7 | 55.2 | 44.7 | 52.6 |
| Ruin Names | 07.8 | 26.9 | 10.1 | 22.4 |
| Formal Fallacies | 50.5 | 54.4 | 51.6 | 53.5 |
| Salient Translation Error Detection | 14.0 | 27.1 | 11.5 | 22.6 |
| CommonsenseQA | 07.4 | 29.1 | 08.8 | 26.9 |
| Coin Flip | 45.2 | 59.4 | 51.1 | 59.7 |

## A.2  Qualitative Analysis

Figure 3 shows an example from CommonsenseQA where GPT-3.5 responses using AO and CoT prompting yield an incorrect answer. The most likely reason for this is that these prompt strategies don't seem to capture all the key points of the input sentence, i.e., the context in the input is based on eyes rather than the overall body. However, this crucial detail is captured when GPT-3.5 is prompted with `AMPLIFY`. We observe that the GPT-3.5 response is correct, and it acknowledges "eyes" as the most important clue in making the correct prediction.

## A.3  Hyper-parameter Analysis

Recall that `AMPLIFY` has two other primary hyper-parameters apart from the rationale template choice discussed in our empirical findings, namely, $s$, which is the size of the few-shot prompt created for LLMs, and $k$, which is the number of most important tokens identified by the post hoc explanation. Table 6 shows the LLM performance variations for different combinations of $(k, s)$. It is important to note that `AMPLIFY` does not have scalability constraints with increasing $s$ and $k$, as `AMPLIFY` computes prompts automatically. This is unlike CoT, where increasing the size of the few-shot prompt would require more human effort to generate relevant chains of thoughts.

## A.4  Impact of BERT as Proxy Model on LLM Performance

Table 7 shows LLM performance when BERT is used as the proxy model in step 1 of `AMPLIFY`. We observe similar trends as those observed for the case of GPT-2, where fine-tuning proxy model provides marginal improvements in general. This indicates that the fine-tuning step could be avoided in most cases to reduce additional computational overhead.

# B  Limitations and Broader Impacts

Our work proposes a new framework, `AMPLIFY`, which focuses on improving the task performance of LLMs by injecting automatically generated rationales. This framework results in the reduction of reliance on processes that require heavy human intervention. These processes, which rely on rationales based on human annotations, often suffer from noise and biases, which may transfer to LLMs during in-context learning. We hope that automated rationale creation will provide a solution to mitigate this problem. While our approach provides significant improvements in model performance, the broader negative impact pertaining to LLMs, such as safety concerns in the form of misinformation[2], social bias[2], hallucination[12], etc., and the massive carbon footprint due to heavy usage of LLMs [17], may still persist even when using our proposed framework. Other than the limitations of LLMs, our framework relies on post hoc explanation methods to create automated rationales; hence, `AMPLIFY` may also inherit widely studied issues with post hoc explanations such as robustness[14], the disagreement problem[13], stability[26], etc.

**Table 6:** This figure shows LLM performance for the different selections of $k$ and $s$ hyper-parameters of AMPLIFY, as denoted by $(k, s)$ for each column. In general, we observe $(k = 7, s = 10)$ achieves the best results for most of the datasets.

| Experiment Tasks | GPT-3 ($k$,$s$) | | | | GPT-3.5 ($k$,$s$) | | | |
|---|---|---|---|---|---|---|---|---|
| | (2, 5) | (5, 5) | (5, 10) | (7, 10) | (2, 5) | (5, 5) | (5, 10) | (7, 10) |
| Snarks | 63.8 | 72.2 | 80.5 | 80.5 | 75.0 | 80.5 | **91.6** | 88.8 |
| Causal Judgment | 52.6 | 57.8 | 60.5 | 60.5 | 65.7 | 73.6 | 76.3 | **76.3** |
| Ruin Names | 64.0 | 75.2 | 76.4 | **78.6** | 73.0 | 75.2 | 77.5 | 77.5 |
| Formal Fallacies | 55.5 | 57.9 | 59.8 | **59.8** | 56.3 | 58.8 | 59.6 | 59.6 |
| Salient Translation Error Detection | 49.7 | 50.2 | 51.2 | 51.2 | 52.7 | 56.2 | 60.8 | **60.8** |
| CommonsenseQA | 72.8 | 73.1 | 73.3 | 73.5 | 76.0 | 76.7 | 77.6 | **77.9** |
| Coin Flip (OOD) | 64.9 | 65.3 | 65.7 | **65.7** | 63.3 | 65.0 | 65.3 | 65.3 |
| All Tasks *(avg)* | 60.4 | 64.5 | 66.7 | 67.1 | 66.0 | 69.4 | **72.6** | 72.3 |

**Table 7:** Few-shot prompting performance of multiple LLMs on the seven datasets when post hoc explanations, which form the rationale in the prompt constructed during step 4 of AMPLIFY, are generated using models with varying degrees of fine-tuning of the proxy model (BERT in this case). Here, "E" represents the number of epochs the proxy model was fine-tuned. "E = 0" indicates that the proxy model was used to generate post hoc explanations without any fine-tuning. The recorded performance in this table represents the percentage of test samples for which the LLM accurately predicted the true label.

| Experiment Tasks | GPT-3 | | | GPT-3.5 | | |
|---|---|---|---|---|---|---|
| | E = 0 | E = 10 | E = 200 | E = 0 | E = 10 | E = 200 |
| Snarks | 66.6 | 72.2 | 72.2 | 80.8 | 80.8 | **88.8** |
| Causal Judgment | 50.0 | 52.6 | 57.8 | 71.0 | 73.6 | **73.6** |
| Ruin Names | 70.7 | 73.0 | **73.0** | 71.9 | 71.9 | 71.9 |
| Formal Fallacies | 56.2 | 56.9 | **58.5** | 56.7 | 56.9 | 57.8 |
| Salient Translation Error Detection | 50.2 | 51.2 | 51.2 | 56.2 | 59.2 | **60.8** |
| CommonsenseQA | 71.3 | 71.8 | 72.4 | 76.1 | 76.5 | **77.4** |
| Coin Flip (OOD) | 65.4 | 65.8 | **65.9** | 63.7 | 64.3 | 65.1 |
| All Tasks *(avg)* | 61.2 | 63.1 | 68.0 | 68.0 | 69.0 | **70.7** |

# C   Additional Experiments

## C.1   Larger Proxy Model: GPT-2-Medium

While we experimented with the fine-tuning of proxy models, it's important to note that this step can be eliminated by using a more capable pretrained proxy model, while still achieving performance gains over baselines. The Table 8 shows that the performance of LLM surpasses the baseline (CoT) when we use gpt2-medium instead of gpt2-small, without any fine-tuning. This demonstrates that fine-tuning of the proxy model is not mandatory. Our motivation to show results for fine-tuning models in the paper is to demonstrate the improvement in LLM performance when the proxy model is further fine tuned.

## C.2   Multi-step Problem Solving : GSM8k

For our experiments, we focused on tasks that require complex language understanding[31], which are also cases where post hoc explanations have been found to be useful in capturing important features, hence providing useful explanations[19]. However, we also experimented with GSM8k (math problem dataset) used in CoT [35] and observed that AMPLIFY outperforms AO but performs lower than CoT, as shown in Table 9.

While we outperform the standard few-shot approach, the underperformance of AMPLIFY when compared to CoT is expected because solving math problems requires multi-step reasoning, a complex function which is beyond what post hoc explanations are designed to explain. We further wish to clarify that we do not present AMPLIFY as a replacement for CoT, but rather as a superior alternative for tasks requiring complex language understanding; these are tasks for which obtaining chains-of-thought through human annotations is exceptionally challenging [31].

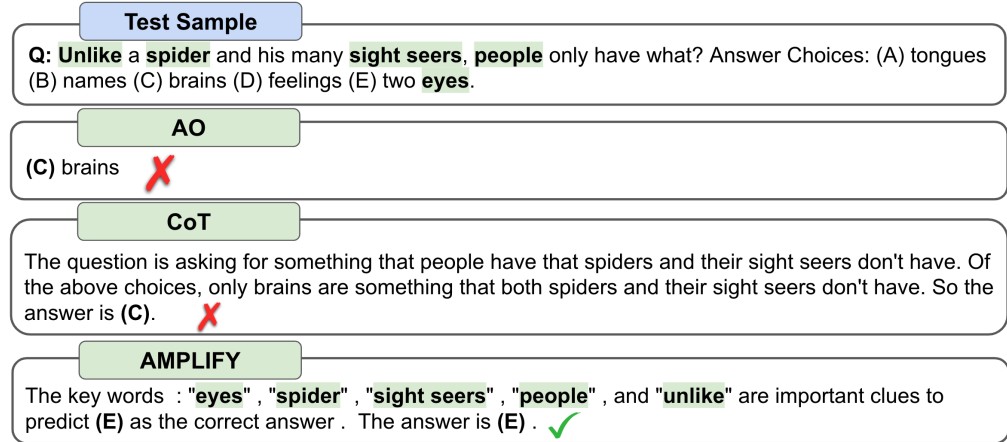

**Figure 3:** This image exemplifies an instance of CommonsenseQA task where standard prompts and CoT produce inaccurate responses. The CoT response fails to take into account the context in the question being related to eyes. In contrast, the response generated by `AMPLIFY` emphasizes this crucial detail.

**Table 8:** The table presents a performance comparison for different models on various experiment tasks. The recorded performance in this table represents the percentage of test samples for which the model accurately predicted the true label.

| Experiment Tasks | AMPLIFY (gpt2-small) | AMPLIFY (gpt2-medium) | CoT |
|---|---|---|---|
| Snarks | 88.8 | **91.6** | 69.4 |
| Causal Judgment | **71.0** | **71.0** | 63.1 |
| Ruin Names | 65.1 | **70.7** | 62.9 |
| Formal Fallacies | 48.3 | **56.0** | 54.6 |
| Salient Translation | 57.7 | **60.8** | 54.7 |
| CommonsenseQA | 71.9 | **75.5** | 75.2 |
| Coin Flip (OOD) | 55.4 | 59.6 | **61.0** |

## C.3 Analysis on Other Big-Bench Hard Tasks

We conducted experiments on some more tasks from Big-Bench Hard [31] and observed similar gains achieved by `AMPLIFY`, shown in Table 10. However, we observed only minimal improvement in the task of word sorting. This is because word sorting requires an understanding of lexical properties over linguistic semantics.

**Table 9:** The table presents a performance comparison for the GSM8k task using three different methods: AO, CoT, and `AMPLIFY` with gpt2-small as the proxy model. The recorded performance in this table represents the percentage of test samples for which the model accurately solved the math problem.

| Experiment Tasks | AO | CoT | `AMPLIFY` (proxy model : gpt2-small) |
|---|---|---|---|
| GSM8k | 22.7 | **43.5** | 27.4 |

**Table 10:** The table presents a performance comparison for the Disambiguation QA, Word Sorting, and Hyperbaton. The recorded performance in this table represents the percentage of test samples for which the model accurately solved the task.

| Experiment Tasks | Random | SOTA | Avg. | Max | AO | CoT | `AMPLIFY` |
|---|---|---|---|---|---|---|---|
| Disambiguation QA | 33.2 | 51.6 | 66.6 | 93.3 | 66.6 | 70.5 | **74.5** |
| Word Sorting | 0 | 33.1 | 62.6 | 100 | 37.8 | 43.1 | **43.6** |
| Hyperbaton | 50.0 | 67.1 | 74.7 | 100 | 68.5 | 77.4 | **79.7** |

