# OpenReview forum: "Post Hoc Explanations of Language Models Can Improve Language Models"
_NeurIPS.cc/2023/Conference — NeurIPS 2023 poster_

### Official Review · Reviewer_mmJT · 2023-06-30

**Soundness:** 3 good
**Presentation:** 4 excellent
**Contribution:** 3 good
**Rating:** 7
**Confidence:** 4

**Summary:**

This paper proposes an alternative prompting framework to Chain-of-thoughts, by using post-hoc explanations from proxy smaller LLMs. Specifically, given a query question and few-shot examples, the method firstly uses a proxy model to get post-hoc explanations on key input words. Then the key input words are combined with few-shot examples to form in-context prompt.

The method is one of the first attempt to utilise post-hoc explanations to boost in-context learning performance. It doesn't require human annotation on reasoning intermediate steps, while outperforming CoT on several challenging tasks from the Big-Bench-Hard benchmark.

**Strengths:**

1. Research on post-hoc explanation have mostly been explored to better understand model prediction. This work potentially opens up a new area of application, as it uses post-hoc explanation to boost in-context learning performance
2. The presentation is clear and easy-to-follow. The author provides informative ablations and comparisons to access the method from multiple aspects, which leads to an optimal settings with sound empirical evidences
3. The improvement with finetuned proxy model, is significant over CoT: 10-25% gain as highlighted in abstract.


**Weaknesses:**

1. Although the improvement over CoT is significant and consistent with fully finetuned proxy model, the improvement becomes less in scale and less consistent with non-finetuned proxy model as shown in Table 2. In fact, if we take a closer look at table 1 & 2, we can see for GPT-3.5 AMPLIFY with non-finetuned proxy model, is worse than CoT on Formal Fallacies(48.3 / 54.6), CommonsenseQA(71.9 / 75.2), Coin Flip (55.4 / 61.0). Besides, such behaviour is not discussed in "Impact of Proxy Model Selection on LLM Performance" paragraph
2. The setting with finetuned proxy model, requires training data for the target task. This breaks the typical assumption of in-context learning where only a handful of annotated examples are available. Besides, it is desirable to show the performance of finetuned proxy model on these tasks (with E=0/10/200), to better assess the benefit of AMPLIFY

**Questions:**

1. When finetuning the proxy model on target task, how many data points did you use? Can we have more information / analysis here with regard to the number of data points used for finetuning vs the number of few-shot examples used to prompt final LLM. It will be good if we can have a fair setting to compare with CoT / AO, where the proxy model is finetuned with the same set of data points as we use to prompt the final LLM.
2. Do you think there are tasks that CoT will work better in principle, like complex / multi-hop reasoning tasks? If so, do you think it's beneficial to combine AMPLIFY with CoT in those cases?

**Limitations:**

The paper has discussed the limitations properly

---

> ### Author Rebuttal · Authors · 2023-08-09
>
> We would like to thank the reviewer for their thoughtful comments, and for recognizing the novelty of our work.  In the subsequent sections, we will address the specific questions and comments raised by the reviewer. Additionally, we are committed to incorporating all of our responses and discussions into the final version of the paper.
>
> >“Although the improvement over CoT is significant and consistent with fully finetuned proxy model..”
>
> As we experimented with the fine-tuning of proxy models, it's important to note that this step can be eliminated by using a more capable pretrained proxy model, yet still achieving performance gains over baselines. The table below shows that the performance of LLM surpasses the baseline when we use gpt2-medium instead of gpt2-small, **without any fine-tuning**. This demonstrates that fine-tuning is not mandatory. We would also like to highlight that our work represents the first pipeline that utilizes post hoc explanation to enhance LLM performance. We have aimed to make it modular to ensure that modifications for improving model performance are easy to implement.
>
> | Experiment Tasks  [LLM : GPT3.5]  | AMPLIFY (gpt2-small) | AMPLIFY(gpt2-medium) | CoT   |
> | ----------------- | ---------- | ----------- | ----- |
> | Snarks| 88.8| **91.6**| 69.4  |
> | CausalJ| 71.0| **71.0** | 63.1  |
> | RuinN | 65.1| **70.7** | 62.9  |
> | Formal Fallacies| 48.3| **56.0**| 54.6  |
> | SalientT | 57.7| **60.8**| 54.7  |
> | CSQA| 71.9 | **75.5**  | 75.2  |
> | Coin Flip (OOD)   | 55.4| 59.6| **61.0**  |
>
>
> Our motivation to show results for fine-tuning models in the paper is to demonstrate the improvement in LLM performance when the proxy model is further fine tuned.
>
> >“The setting with finetuned proxy model, requires training data…”
>
> Thank you for your comment.  Appendix “A.1 Proxy Model Task Performance” shows the proxy model performance with E=0 and E=200. Even with E=200 finetuning, the proxy models' performance is far below compared to performance achieved by AMPLIFY. There are two reasons behind this low performance of proxy models : (1) these tasks are extremely challenging for such small models [1], (2) The train samples used for fine-tuning weren’t enough to achieve significant performance improvements. For instance, we used 87, 91, 215, 479 samples to fine-tune for Snarks, Causal Judgment, Salient Translation, and Ruin Names.
>
> Furthermore, we would like to emphasize that performance measurements with fine-tuning were provided for the sole purpose of understanding changes in performance if the proxy model is fine-tuned with training samples. As demonstrated in the table responding to the previous comment, the fine-tuning step can be entirely eliminated with the use of a slightly more effective model. We will include this detail in the final draft for more clarity.
>
>
> >“Can we have more information / analysis here with regard to the number of data points used for finetuning vs the number of few-shot examples used to prompt final LLM.
>
> In order to create the validation set, we randomly selected 40% of the total samples from the train set for each dataset. We used s=10 (number of few-shot examples) and k=5 (# top-k tokens)  as the default hyper-parameters for results shown in all the tables. We provide results on other hyper-parameter settings in Table 6 of Appendix A.3.  We will include this detail in the final draft.
>
>
> >”It will be good if we can have a fair setting ..”
>
> In the table below, we show performance gains when the proxy model (gpt2-small) is fine-tuned with the same samples chosen for the final LLM. We observe the performance to be very similar to what we obtained after fine-tuning with all train data for E=10.
>
> | Experiment Tasks [LLM : GPT3.5] |  AO   | CoT  | AMPLIFY (gpt2-small, E = 50) |   AMPLIFY (reported in Table 1)    |
> |------------------|------|------|----------|-----------|
> |     CausalJ       | 57.8 | 63.1 |   73.6   | **76.3**  |
> |      RuinN        | 69.6 | 62.9 |    68.5  |   **77.5**    |
> |     SalientT      | 43.2 | 54.7 |    55.2  | **60.8**  |
> |      CSQA        | 75.7 | 75.2 |   76.0   | **77.9**  |
> |  Coin Flip(OOD)   | 52.9 | 61.0 |   61.7   |   **65.3**    |
>
> >“Do you think there are tasks that CoT will work better in principle…”
>
> Thank you for your comment. For our experiments, we focused on tasks that require complex language understanding [2], which are also cases where post hoc explanations have been found to be useful in capturing important features, hence providing useful explanations [3].
>
>
> However, we also experimented with GSM8k (math problem dataset) used in CoT and observed that AMPLIFY outperforms AO but performs worse than CoT. shown in table below.
>
>
> | Experiment Tasks | AO | CoT | AMPLIFY (proxy model : gpt2-small) |
> | --- | --- | --- | --- |
> | GSM8k | 22.7 | **43.5** | 27.4 |
>
>
> While we outperform the standard few-shot approach, the underperformance of AMPLIFY when compared to CoT is expected because solving math problems requires multi-step reasoning, a complex function which is beyond what post hoc explanations are designed to explain. We further wish to clarify that we do not present AMPLIFY as a replacement for CoT, but rather as a superior alternative for tasks requiring complex language understanding; these are tasks for which obtaining chains-of-thought through human annotations is exceptionally challenging [2].
>
> The goal of our approach is to generate explanations without any dependence on human annotations. However, a combination of CoT and AMPLIFY could lead to further gains. This will require more analysis and could be an interesting area for future research.
>
> **References**
>
> [1]​​ Srivastava, Aarohi, et al. Beyond the imitation game: Quantifying and extrapolating the capabilities of language models.(2022)
>
> [2] Suzgun, Mirac, et al. Challenging big-bench tasks and whether chain-of-thought can solve them.(2022)
>
> [3]  Madsen, Andreas, et. al. Post-hoc interpretability for neural nlp: A survey. (2021)

---

> > ### Comment · Reviewer_mmJT · 2023-08-15
> > **Response to Author**
> >
> > Thank you for the thoughtful and detailed response!
> >
> > I agree that AMPLIFY will work better on tasks requiring "complex language understanding" and it's expected that for multi-step reasoning tasks like GSM8K, AMPLIFY will be less effective to CoT. It's a good practice that the authors share those results, which help us readers to understand better of the applicability of AMPLIFY and have a clearer understanding of its contribution.
> >
> > With the author's response, I perceive AMPLIFY as one of the first attempt on using post-hoc explanation for improving generation performance on tasks requiring "complex language understanding". It serves as a nice alternative prompting method to CoT as it doesn't require any manual design of CoT prompts. I will confidently keep my recommendation score unchanged based on this perception.

---

> > > ### Author Response · Authors · 2023-08-21
> > >
> > > Thank you very much for reviewing our response! We will include the additional results in the final draft.

---

### Official Review · Reviewer_4Npw · 2023-07-02

**Soundness:** 3 good
**Presentation:** 3 good
**Contribution:** 3 good
**Rating:** 6
**Confidence:** 4

**Summary:**

This paper demonstrates how post hoc explanations through a small LM could assist the performance of LLMs. The process is divided into 4 steps: Proxy Model Selection, Few-shot sample selection, compute explanations, and formatting prompts for LLMs. This technique automatically generates few-shot demonstrations, reducing human-annotation for few-shot in-context learning.

**Strengths:**

- The method is sound and novel.
- The paper contains extensive ablation experiments.
- The improvement for some tasks is dramatic. (e.g. snarks)


**Weaknesses:**

- This paper does not compare the performance with other baselines such as Auto-CoT that automatically generate demonstrations for in-context learning.
Zhang et al (2022) Automatic Chain of Thought Prompting in Large Language Models

- Although it is stated that the experimentation is specifically carried out on datasets aimed at evaluating complex linguistic understanding concepts, it omits datasets such as DisambiguationQA, Hyperbaton, and Word Soriting of Big-Bench-Hard benchmark.

- When GPT-2 is not fine-tuned (E=0) in Table 2, AMPLIFY outperforms both AO and CoT on 3 out of 7 datasets for both GPT-3 and GPT-3.5, indicating that fine-tuning process is crucial. Also, the performance gets worse compared to AO for datasets such as formal fallacies, commonsenseQA, and ruin names for GPT-3.5.



**Questions:**

- When fine-tuning GPT-2, what training datasets are used? As far as I know, tasks of Big-Bench-Hard do not contain training instances.

- Do you expect that AMPLIFY would be still effective for models with less than 100B parameters? Also, do you think the effect of AMPLIFY could generalize to open-source LLMs?

- What is the default s value (number of shots) for GPT-3, GPT-3.5 result of Table 1?

**Limitations:**

The authors adequately addressed the limitations and potential negative societal impact of their work.

---

> ### Author Rebuttal · Authors · 2023-08-09
>
> We thank the reviewer for the valuable thoughts and suggestions. We appreciate the reviewer's acknowledgement of the novelty and analysis provided in our work. In the subsequent sections, we will address the specific questions and comments raised by the reviewer. Additionally, we are committed to incorporating all of our responses and discussions into the final version of the paper.
>
> >“This paper does not compare the performance with other baselines..”
>
> Thank you for your comment. We experimented with the specific baseline suggested in the comment and found that our method performs better than Auto-CoT, as shown in the table below. We believe the reason behind the under-performance of Auto-CoT (and similar self-reasoning based methods) is that the reasoning generated by LLMs is not reliable and has a higher chance of being incorrect, which has also been shown in several research works [1, 2, 3].
> | Experiment Tasks [LLM : GPT3.5] |  AO   | CoT  | Auto-CoT |   AMPLIFY    |
> |------------------|------|------|----------|-----------|
> |     Causal Judgment       | 57.8 | 63.1 |   63.1   | **76.3**  |
> |      Ruin Names        | 69.6 | 62.9 |   67.4   |   **77.5**    |
> |     Salient Translation      | 43.2 | 54.7 |   53.2   | **60.8**  |
> |      CommonsenseQA        | 75.7 | 75.2 |   74.6   | **77.9**  |
> |  Coin Flip(OOD)   | 52.9 | 61.0 |   62.9   |   **65.3**    |
>
> >“Although it is stated that the experimentation is specifically carried out on datasets aimed at ..”
>
> Thank you for your comment. We conducted experiments on the suggested datasets and observed similar gains achieved by AMPLIFY. However, we observed only minimal improvement in the task of word sorting. This is because word sorting requires an understanding of lexical properties over linguistic semantics.
> | Experiment Tasks   [LLM : GPT3.5]  | Random | SOTA  | Avg.  | Max   | AO    | CoT   | AMPLIFY |
> |--------------------|--------|-------|-------|-------|-------|-------|-------|
> | Disambiguation QA  | 33.2   | 51.6  | 66.6  | 93.3  | 66.6  | 70.5  | **74.5**  |
> | Word Sorting       | 0      | 33.1  | 62.6  | 100   | 37.8  | 43.1  | **43.6**  |
> | Hyperbaton         | 50.0   | 67.1  | 74.7  | 100   | 68.5  | 77.4  | **79.7**  |
>
> >”When GPT-2 is not fine-tuned (E=0) in Table 2, AMPLIFY outperforms…”
>
> While we experimented with the fine-tuning of proxy models, it's important to note that this step can be eliminated by using a more capable pretrained proxy model, and still achieve performance gains over baselines. The table below shows that the performance of LLM surpasses the baseline when we use gpt2-medium instead of gpt2-small, without any fine-tuning. This demonstrates that fine-tuning is not mandatory.
> | Experiment Tasks  [LLM : GPT3.5]  | AMPLIFY (gpt2-small) | AMPLIFY(gpt2-medium) | CoT   |
> | ----------------- | ---------- | ----------- | ----- |
> | Snarks           | 88.8       | **91.6**        | 69.4  |
> | Causal Judgment           | 71.0       | **71.0**        | 63.1  |
> | Ruin Names             | 65.1       | **70.7**        | 62.9  |
> | Formal Fallacies          | 48.3       | **56.0**        | 54.6  |
> | Salient Translation         | 57.7       | **60.8**        | 54.7  |
> | CommonsenseQA             | 71.9       | **75.5**        | 75.2  |
> | Coin Flip (OOD)   | 55.4       | 59.6        | **61.0**  |
>
> Our motivation to show results for fine-tuning models in the paper is to demonstrate the improvement in LLM performance when the proxy model is further fine tuned.
>
> >“When fine-tuning GPT-2, what training datasets are used?..”
>
> Generally, the Big-Bench-Hard datasets have been used in both settings : (1) evaluation only and (2) train/test split versions. For our experiments, we used the version that provides a train-test split, which is publicly hosted on Hugging Face.
>
>
> >“Do you expect that AMPLIFY would be still effective for models…”
>
> In our work, we analyzed models with a parameter size of >100Bn for two reasons: (1) These are models that have demonstrated emergent abilities in performing extremely challenging tasks [4], and (2) to provide a fair comparison against CoT, which performs poorly on LLMs with a parameter size smaller than 100Bn. We ran a quick experiment with Alpaca7B, an open-source LLM, on the Snarks dataset and found that AMPLIFY performs at 44.4, compared to 38.8 (AO) and 41.6 (CoT). This suggests that AMPLIFY may also be used for smaller models (<100Bn), but a deeper analysis is required, which could be a potential area for future work.
>
>
> >“What is the default s value (number of shots) for .."
>
> We used s=10 and k=5 as the default hyper-parameters for results shown in all the tables. We provide results on other hyper-parameter settings in Table 6 of Appendix A.3.
>
>
> **References**
>
> [1] Turpin, M., Michael, J., Perez, E., & Bowman, S. R. (2023). Language Models Don't Always Say What They Think: Unfaithful Explanations in Chain-of-Thought Prompting. arXiv preprint arXiv:2305.04388.
>
> [2] Lanham, Tamera, et al. "Measuring Faithfulness in Chain-of-Thought Reasoning." arXiv preprint arXiv:2307.13702 (2023).
>
> [3] Zhao, Ruochen, et al. "Verify-and-edit: A knowledge-enhanced chain-of-thought framework." arXiv preprint arXiv:2305.03268 (2023).
>
> [4] Wei, Jason, et al. "Emergent abilities of large language models." arXiv preprint arXiv:2206.07682 (2022).

---

> > ### Comment · Reviewer_4Npw · 2023-08-21
> >
> > Thank you for your response and sharing of additional results. I think the additional results would give more valuable insights to the reader. I will keep my score as "weak accept".

---

> > > ### Author Response · Authors · 2023-08-21
> > >
> > > Thank you very much for reviewing our response! We will include the additional results in the final draft.

---

### Official Review · Reviewer_LsYX · 2023-07-04

**Soundness:** 2 fair
**Presentation:** 2 fair
**Contribution:** 2 fair
**Rating:** 5
**Confidence:** 4

**Summary:**

The paper presents AMPLIFY, an approach that uses post-hoc explanations from a proxy model to improve the prompting performance of large language models. For a given dataset, the approach assumes access to a set of labeled validation data that is used for crafting a prompt. First, the approach selects k examples that are misclassified by LLMs and exhibit high misclassification confidence scores. Next, post-hoc explanation techniques are used to find the most important input features of these selected examples, which are later used to construct template-based rationales. The rationales are used in the final prompts following the chain-of-thought paradigm.

The paper conducts experiments on 7 datasets from Big-Bench-Hard. Experimental results suggest good improvements over the few-shot answer-only prompting (AO) baseline and the few-shot CoT baseline. The paper also includes analyses suggesting the importance of example selection strategies as well as the impacts of using different proxy models and explanation techniques.


**Strengths:**

The paper studies an interesting problem that focuses on using post-hoc explanations for improving LLMs’ performance, which is relatively new.

The experiments cover 7 datasets and show performance improvements compared to few-shot baselines.

The paper provides some useful analyses, especially the analysis of the example selection strategies.

The paper is well-written and easy to follow.


**Weaknesses:**

1: The major comparison is unfair in some way

The proposed approach uses a validation set (the size of the validation set is also not mentioned in the paper, nor in the appendix) and selects examples from the validation set. By contrast, standard prompting and chain-of-thought prompting typically only use few-shot examples. So part of the improvements might also be attributed to the use of more data in addition to the prompting method proposed in this paper (see weakness 2). In addition, the paper does not compare against approaches that can also use the validation set instead of just a few-shot examples.

2: However, the experiments do not clearly suggest the effectiveness of using post-hoc explanations

The proposed approach uses the LLM itself + smaller proxy models to find more informative examples to be used in the prompts (the selection strategy). This active selection strategy does not rely on using post-hoc explanations and can be applied to answer-only prompting or chain-of-thought prompting as well. It would be good to provide AO and CoT performance using the selected examples as well to give a better understanding of the effectiveness of using better examples versus the effectiveness of using post-hoc explanations in prompts.

The paper does provide analysis of the impacts of using different selection strategies. As shown in Table 3, using random examples, the performance of AMPLIFY is only 59.3 on GPT3 and 62.0 on GPT-3.5; at the same time, according to Table 1, CoT performance is 58.0 on GPT3 and 62.9 on GPT-3.5. IIUC, this suggests CoT and AMPLIFY is comparable; and selecting better examples contribute the majority of the performance improvements.

While it seems that most of the improvements come from the selection strategy, the paper does not compare against other approaches that involve active annotating examples for in-context learning (e.g., Su et al. (2022); Diao et al. (2023)). Also, this proposed selective annotation strategy does not distinguish itself from existing work, which uses similar confidence-based ways to actively annotate examples (Su et al., 2022; Diao et al., 2023).

Selective Annotation Makes Language Models Better Few-Shot Learners (Su et al., 2022)
Active Prompting with Chain-of-Thought for Large Language Models (Diao et al., 2023)

3: The paper also misses some important details on the approach and the baseline
Regarding the proposed approach:
* When getting the post-hoc explanations. Does it explain the ground truth label or the predicted label? (it is not clearly explained in line 172)
* How does the method get the initial predictions, in order to determine the misclassified examples? Is it obtained using few-shot AO or few-shot CoT?
* It seems no information regarding the size of the validation set is provided.

Regarding the baselines:
* It seems the CoT baselines and AO baselines use few-shot prompting. How is the shot selected? Are they randomly selected? How many shots are used?
* How are the CoTs for the CoT baseline written? Based on line 129, the CoTs are taken from Wei et al. (2022), but the original CoT paper does not include prompts for these datasets. Are they from Suzgun et al. (2022)?

In addition, it would be also helpful if the appendix can include some prompt examples.

4: The paper uses a subset from big-bench-hard that mainly focuses on understanding linguistic concepts.

On these datasets, CoT typically leads to no improvements or minor improvements, compared to AO. While it is understandable that the paper chooses to focus on understanding linguistics concepts, it would still be beneficial to benchmark the effectiveness of this approach on a broader range of multi-step reasoning tasks (e.g., GSM) where CoT shows substantial improvements over AO. And on these tasks, using rationales consisting of just top-k words may be less effective.




**Questions:**

See weakness

**Limitations:**

The paper discusses the limitations in appendix.

---

> ### Author Rebuttal · Authors · 2023-08-09
>
> We thank the reviewer for their insightful comments and for acknowledging the novelty and insights presented in our work. In the subsequent sections, we will address the specific questions and comments raised by the reviewer. Furthermore, we intend to incorporate all our responses and discussions into the final version of the paper.
>
> **Impact of Post Hoc Explanations**
>
> In the table below, we show that the performance of AO with the samples selected using step-2 of AMPLIFY, denoted by AO-AMPLIFY, is worse than AMPLIFY which suggests that it is not only the sample selection but also the explanations for each sample that helps in improving LLM performance.
> |Experiment Tasks [LLM : GPT3.5]|AO|AO-AMPLIFY|AMPLIFY|
> |------------------|---|----------|--------|
> |CausalJ|57.8|63.1|**76.3**|
> |RuinN|69.6|73.0|**77.5**|
> |SalientT|43.2|48.7|**60.8**|
> |CSQA|75.7|75.1|**77.9**|
> |CoinFlip|52.9|55.2|**65.3**|
>
> It is important to note that we cannot experiment with CoT using samples chosen by AMPLIFY because CoT has hard-coded prompts which were created with human assistance for only a fixed set of samples for which their method performed the best [2].
>
> **Other Baselines**
>
> We experimented with two other baselines Auto-CoT and Vote-k (Su et al., 2022), which requires additional data to create the prompt, and found that our method performs better, as shown in the table below. We believe the reason behind the under-performance of Auto-CoT (and similar self-reasoning based methods) is that the reasoning generated by LLMs is not reliable and has a higher chance of being incorrect, which has also been shown in several research works [2,4]. Performance for Vote-k remained close to AO because it doesn’t include additional explanations for the few-shot samples in the prompt.
> |Experiment Tasks [LLM : GPT3.5]|AO|CoT|Auto-CoT|Vote-k|AMPLIFY|
> |---------------------------------|-------|------|----------|-------|--------------|
> |CausalJ|57.8|63.1|63.1|55.2|**76.3**|
> |RuinN|69.6|62.9|67.4|64.0|**77.5**|
> |SalientT|43.2|54.7|53.2|47.7|**60.8**|
> |CSQA|75.7|75.2|74.6|73.9|**77.9**|
> |CoinFlip|52.9|61.0|62.9|54.7|**65.3**|
>
> We cannot make a fair comparison with Diao et al. (2023) because their work involves human-assisted annotations for datasets that are different from those requiring complex language understanding, which are the tasks we focused on in our study. In contrast to the methods suggested by Su et al. (2022) and Diao et al. (2023), our approach eliminates the need for any annotation step, thereby removing any dependence on human assistance.
>
> >“It seems no information..”
>
> Thank you for your comment.  In order to create the validation set, we randomly selected 40% of the total samples from the train set for each dataset. We will include this detail in the final draft.
>
> >“Does it explain the ground truth label..”
>
> Post hoc explanations are generated with respect to the ground truth label. The intuition behind this is to obtain relevant corrective signals, in the form of the tokens important for the prediction of the ground truth label, that could assist LLM to make the correct prediction. We have mentioned this detail in the figure caption and on lines 174-175, but we will elaborate on it further in the final draft for more clarity.
>
> >“How does the method get the initial predictions..”
>
> We used zero-shot prompting to obtain misclassified examples, i.e., the prompt only contains the sample from the validation set for the LLM to predict the response. We adopted this setting for two reasons: (1) to acquire as many misclassified examples as possible, thereby obtaining sufficient post hoc explanations for the LLM to make corrections, and (2) to maintain consistency with the post hoc explanation generation step. The post hoc explanation for each sample is computed on a zero-shot prompt, ensuring that the resulting explanation consists solely of important tokens from the input sample, rather than tokens from other samples as can occur with a few-shot prompt.
>
> >“It seems the CoT baselines and AO baselines..”
>
> For our experiments, we used the same few shot prompts provided in [1,2] for both CoT and AO baselines.
>
> >“How are the CoTs for the CoT baseline written? ..”
>
> Yes, the chain-of-thoughts for tasks other than the one experimented in [1] is taken from [2].
>
> >“ appendix can include..”
>
> Thank you for your comment. We will revise the appendix with prompt examples.
>
> >“it would still be beneficial to benchmark the effectiveness…”
>
> Thank you for your comment. For our experiments, we focused on tasks that require complex language understanding [2], which are also cases where post hoc explanations have been found to be useful in capturing important features, hence providing useful explanations [3].
> However, we also experimented with GSM8k (math problem dataset) used in CoT and observed that AMPLIFY outperforms AO but performs worse than CoT. shown in the table below.
> | Experiment Tasks | AO | CoT | AMPLIFY (gpt2-small) |
> | --- | --- | --- | --- |
> | GSM8k | 22.7 | **43.5** | 27.4 |
>
> While we outperform the standard few-shot approach, the underperformance of AMPLIFY when compared to CoT is expected because solving math problems requires multi-step reasoning, a complex function which is beyond what post hoc explanations are designed to explain. We further wish to clarify that we do not present AMPLIFY as a replacement for CoT, but rather as a superior alternative for tasks requiring complex language understanding; these are tasks for which obtaining chains-of-thought through human annotations is exceptionally challenging [2].
>
> **References**
>
> [1] Wei, Jason, et al.Chain-of-thought prompting elicits reasoning in large language models.(2022)
>
> [2] Suzgun, Mirac, et al.Challenging big-bench tasks and whether chain-of-thought can solve them.(2022)
>
> [3]  Madsen, Andreas, et. al. Post-hoc interpretability for neural nlp: A survey.(2021)
>
> [4] Zhang, Zhuosheng, et al. Automatic chain of thought prompting in large language models(2022)

---

> > ### Comment · Reviewer_LsYX · 2023-08-11
> >
> > Thank you for the response and clarification. I appreciate the added results for isolating the impacts of selecting better examples. I've raised by score.

---

> > > ### Author Response · Authors · 2023-08-21
> > >
> > > Thank you for taking the time to review our response. We greatly appreciate your recognition of our work and for increasing the score.

---

### Official Review · Reviewer_sqcU · 2023-07-05

**Soundness:** 3 good
**Presentation:** 3 good
**Contribution:** 2 fair
**Rating:** 5
**Confidence:** 4

**Summary:**

This paper proposes AMPLIFY framework that leverages post-hoc explanations to generate rationales automatically for chain-of-thought prompting. The framework consists of four stages: (1) adopt a light-weight model as the proxy to compute explanations, (2) select few-shot samples misclassified by LLM, (3) use attribution scores to identify the important words as rationales, (4) prompt LLM for predictions. Experimental results show that AMPLIFY outperforms the previous methods by a large margin across diverse tasks.


**Strengths:**

1. The paper is well-written and well-motivated.
2. The idea of leveraging the supervision from the small models to improve LLM is novel.
3. The results on seven tasks demonstrate the effectiveness of the proposed method.

**Weaknesses:**

Overall, the proposed method seems too heavy and its costs outweigh the performance benefits.
1. It requires an extra proxy model and fine-tuning on every target task in most cases, which can be impractical and laborious.
2. The second stage selects misclassified samples from the entire validation set and conducts filtering with a pre-defined metric, which breaks the constraints of "few-shot".
3. The third stage selects the top-k most important words to construct the rationale, which does not take into account the interactions between words that affect the model predictions.
4. The datasets they evaluated on are limited. The authors do not compare their method with CoT on more complex tasks like math problems and multi-hop reasoning.

In contrast, CoT performs well under few-shot settings and only requires a few human-annotated rationales without any extra proxies and training costs.

**Questions:**

1. Experiments show that fine-tuning is necessary in most cases (compared with "Random" baseline). What is the minimum size of a validation set to ensure that the explanations provided by the proxy model are reliable? (If it is too small, the proxy model is prone to overfitting.)
2. I'm curious about the performance of the proposed method on math problems compared with CoT. Is it general enough to improve the performance on various tasks?

**Limitations:**

The authors adequately address the limitations of their work.

---

> ### Author Rebuttal · Authors · 2023-08-08
>
> We are grateful to the reviewer for their insightful comments and suggestions. We are pleased to know they acknowledge our novel approach of using smaller models to augment the decision-making capabilities of larger ones. In the following sections, we address specific questions and comments raised by the reviewer. Furthermore, we intend to incorporate all our responses and discussions into the final version of the paper.
>
> >“requires an extra proxy model and fine-tuning..”
>
> While we experimented with the fine-tuning of proxy models, it's important to note that this step can be eliminated by using a more capable pretrained proxy model, while still achieving performance gains over baselines. The table below shows that the performance of LLM surpasses the baseline when we use gpt2-medium instead of gpt2-small, **without any fine-tuning**. This demonstrates that fine-tuning is not mandatory. Our motivation to show results for fine-tuning models in the paper is to demonstrate the improvement in LLM performance when the proxy model is further fine tuned.
> | Experiment Tasks  [LLM : GPT3.5]  | AMPLIFY (proxy model :gpt2-small) | AMPLIFY(proxy model :gpt2-medium) | CoT   |
> | ----------------- | ---------- | ----------- | ----- |
> | Snarks | 88.8| **91.6** | 69.4|
> | Causal Judgment| 71.0  | **71.0** | 63.1  |
> | Ruin Names| 65.1  | **70.7**    | 62.9  |
> | Formal Fallacies  | 48.3  | **56.0**   | 54.6  |
> | Salient Translation   | 57.7  | **60.8**  | 54.7  |
> | CSQA    | 71.9  | **75.5** | 75.2  |
> | Coin Flip (OOD) | 55.4  | 59.6  | **61.0**  |
>
> >“third stage selects the top-k most important words…”
>
> One of the benefits of AMPLIFY is that we do not need to necessarily account for the interaction between the words in the explanation, as the LLM already sees them in context. To confirm this hypothesis, we have conducted an additional experiment: We have now added a new variant of AMPLIFY that highlights the most influential keywords in the example rather than listing them or their interactions separately. This variant still results in significant improvements over baselines, as was observed in Table 1 of the paper. It requires a shorter context length and provides further empirical evidence supporting this hypothesis. This advantage of not providing word interactions also makes AMPLIFY more computationally efficient since providing word interactions in the prompt is a non-trivial and computationally expensive process [5,6,7].
>
> >“datasets they evaluated on are limited...”
>
> Thank you for your comment. For our experiments, we focused on tasks that require complex language understanding [1], which are also cases where post hoc explanations have been found to be useful in capturing important features, hence providing useful explanations [2].  However, we also experimented with GSM8k (math problem dataset) used in CoT and observed that AMPLIFY outperforms AO but performs worse than CoT, shown in the table below.
>
> | Experiment Tasks | AO | CoT | AMPLIFY (proxy model : gpt2-small) |
> | --- | --- | --- | --- |
> | GSM8k | 22.7 | **43.5** | 27.4 |
>
> While we outperform the standard few-shot approach, the underperformance of AMPLIFY when compared to CoT is expected because solving math problems requires multi-step reasoning, a complex function which is beyond what post hoc explanations are designed to explain. We further wish to clarify that we do not present AMPLIFY as a replacement for CoT, but rather as a superior alternative for tasks requiring complex language understanding; these are tasks for which obtaining chains-of-thought through human annotations is exceptionally challenging [1].
>
> >“The second stage selects misclassified samples from...”
>
> Thank you for your comment. We observed no change in performance when only the misclassified samples from the proxy model were selected, a common sample selection strategy in few-shot learning paradigms used in several other research works [3,4]. We believe the reason behind no change in performance is due to the fact that the misclassifications made by the proxy model generally encompass all the misclassified samples from the LLM, covering all samples that can provide corrective signals to the LLM. We will include this detail in the final draft for clarity.
>
> >“What is the minimum size of a validation set.. ”
>
> Thank you for your comment. To create the validation set, we randomly selected 40% of the total samples from the train set for each dataset. According to the accuracies of the fine-tuned proxy models presented in Table 5 of Appendix A.1, the accuracy after fine-tuning continues to be worse than the performance achieved by AMPLIFY. Therefore, it can be concluded that the proxy models have not overfit.
>
> >“the proposed method seems too heavy..”
>
> We address this comment in more detail in the global comment under “Trade-Off between Computational Cost and Performance”.
>
>
> **References**
>
> [1] Suzgun, M., et.al. Challenging big-bench tasks and whether chain-of-thought can solve them.
>
> [2] Madsen, Andreas, et. al "Post-hoc interpretability for neural nlp: A survey."
>
> [3] Zhang, Zhuosheng, et al. "Automatic chain of thought prompting in large language models."
>
> [4] Chang, Ernie, et al. "The selectgen challenge: Finding the best training samples for few-shot neural text generation."
>
> [5] Byrd, Roy J.,et. al. Identifying and extracting relations in text. na,
>
> [6] Agichtein, Eugene, et.al "Snowball: Extracting relations from large plain-text collections."
>
> [7] Bunescu, Razvan C., et. al "Extracting relations from text: From word sequences to dependency paths."

---

> > ### Comment · Reviewer_sqcU · 2023-08-18
> >
> > Thanks for the clarification. I will raise my score.

---

> > > ### Author Response · Authors · 2023-08-21
> > >
> > > Thank you for taking the time to review our response. We greatly appreciate your recognition of our work and for increasing the score.

---

### Author Rebuttal · Authors · 2023-08-09

We are grateful to the reviewers for their valuable and insightful comments. It is appreciated that the reviewers found our work to be well-written and well-motivated, and considered our analysis to be novel and insightful in improving LLM performance. In this section, we would like to address the common concerns raised by the reviewers.

**Trade-Off between Computational Cost and Performance**

Our research is focused on enhancing the performance of Large Language Models (LLMs) in complex language understanding tasks. These tasks include causal judgment and sarcasm detection, among others, where creating an appropriate Chain-of-Thought (CoT) is extremely challenging. This weakness is studied in more detail in [1, 2, 3, 4]. We propose a method that provides post hoc explanations to assist LLMs in identifying key components of the input for accurate decision-making. This method offers significant performance improvements compared to the standard few-shot learning (AO) and CoT. Our method only requires the computation of post hoc explanations for a smaller proxy model ($\sim$0.1Bn params). This is thousands of times less computationally demanding than a single inference with an LLM ($\sim$175Bn params) and can be executed without additional GPU support. This makes the cost significantly lower compared to the substantial performance improvements in complex language understanding tasks. We hope this clarifies the trade-offs between computational cost and task performance resulting from our approach.


**Importance of Fine-tuning Proxy Model**

While we experimented with the fine-tuning of proxy models, it's important to note that this step can be eliminated by using a more capable pretrained proxy model, while still achieving performance gains over baselines. The table below shows that the performance of LLM surpasses the baseline (CoT) when we use gpt2-medium instead of gpt2-small, **without any fine-tuning**. This demonstrates that fine-tuning of the proxy model is not mandatory. Our motivation to show results for fine-tuning models in the paper is to demonstrate the improvement in LLM performance when the proxy model is further fine tuned.

| Experiment Tasks  [LLM : GPT3.5]  | AMPLIFY (gpt2-small) | AMPLIFY(gpt2-medium) | CoT   |
| ----------------- | ---------- | ----------- | ----- |
| Snarks           | 88.8       | **91.6**        | 69.4  |
| Causal Judgment           | 71.0       | **71.0**        | 63.1  |
| Ruin Names             | 65.1       | **70.7**        | 62.9  |
| Formal Fallacies          | 48.3       | **56.0**        | 54.6  |
| Salient Translation         | 57.7       | **60.8**        | 54.7  |
| CommonsenseQA             | 71.9       | **75.5**        | 75.2  |
| Coin Flip (OOD)   | 55.4       | 59.6        | **61.0**  |


 We would also like to highlight that our work represents the first pipeline that utilizes post hoc explanation to enhance LLM performance. We have aimed to make it modular to ensure that modifications for improving model performance are easy to implement. The modularity of our method also helps users to make most out of their computational resources.


**Other Baselines**

Reviewers suggested few baseline methods other than the ones used in our experiments. Hence, we experimented with these baselines, i.e, Auto-CoT[2] and Vote-k[5], and found that our method performs better, as shown in the table below. We believe the reason behind the under-performance of Auto-CoT (and similar self-reasoning based methods) is that the reasoning generated by LLMs is not reliable and has a higher chance of being incorrect, which has also been shown in several research works [1,2,4]. Performance for Vote-k remained close to AO because it doesn’t include additional explanations for the few-shot samples in the prompt.


| Experiment Tasks [LLM : GPT3.5] |  AO   | CoT  | Auto-CoT |Vote-k |    AMPLIFY   |
|---------------------------------|-------|------|----------|-------|--------------|
| Causal Judgment                         | 57.8  | 63.1 |   63.1   | 55.2  | **76.3**    |
| Ruin Names                           | 69.6  | 62.9 |   67.4   | 64.0  | **77.5**    |
| Salient Translation                       | 43.2  | 54.7 |   53.2   | 47.7  | **60.8**    |
| CommonsenseQA                            | 75.7  | 75.2 |   74.6   | 73.9  | **77.9**    |
| Coin Flip (OOD)                        | 52.9  | 61.0 |   62.9   | 54.7  | **65.3**    |



 **References**


[1] Suzgun, M., et.al. (2022). Challenging big-bench tasks and whether chain-of-thought can solve them. arXiv preprint arXiv:2210.09261.

[2] Zhang, Zhuosheng, et al. "Automatic chain of thought prompting in large language models." arXiv preprint arXiv:2210.03493 (2022).

[3] Turpin, M.,et. al(2023). Language Models Don't Always Say What They Think: Unfaithful Explanations in Chain-of-Thought Prompting. arXiv preprint arXiv:2305.04388.

[4] Lanham, Tamera, et al. "Measuring Faithfulness in Chain-of-Thought Reasoning." arXiv preprint arXiv:2307.13702 (2023).

[5] Su, Hongjin, et al. "Selective annotation makes language models better few-shot learners." arXiv preprint arXiv:2209.01975 (2022).

---

### Decision · Program_Chairs · 2023-09-21

**Decision:**

Accept (poster)

**Comment:**

The paper proposes a novel method that creates post-hoc explanation (attribution scores) of language models, which capture the influence of each example input on the prediction. The method is effective, improving accuracy by 10-25% on a variety of tasks and compliments Chain of Thoughts method by enhancing performance on those tasks that CoT does not do well. While there were a few concerns by the reviewers, such as higher inference cost (than CoT) and lack of comparison experiments with some baselines, the authors have addressed many of these issues during rebuttal period. Overall, the paper might need some work on clarifying its contents in the final revision but the method it proposes gives a new perspective on demystifying how language models generalize well and also have meaningful practical applications, so I recommend it to be included in the proceedings of NeurIPS 2023.